# Intelligent Healthcare System Using Mathematical Model and Simulated Annealing to Hide Patients Data in the Low-Frequency Amplitude of ECG Signals

**DOI:** 10.3390/s22218341

**Published:** 2022-10-30

**Authors:** Chih-Yu Hsu, Chih-Cheng Chen, Chun-You Liu, Shuo-Tsung Chen, Shu-Yi Tu

**Affiliations:** 1School of Transportation, Fujian University of Technology, Fuzhou 350118, China; 2Department of Automatic Control Engineering, Feng Chia University, Taichung 40724, Taiwan; 3Department of Aeronautical Engineering, Chaoyang University of Technology, Taichung 413310, Taiwan; 4Ph.D. Program of Electrical and Communications Engineering, Feng Chia University, Taichung 40724, Taiwan; 5Department of Applied Mathematics, Tunghai University, Taichung 40704, Taiwan; 6Department of Mathematics, University of Michigan, Flint, MI 48502, USA

**Keywords:** healthcare system, confidential data, low-frequency, wavelet-domain, nonlinear model, simulated annealing

## Abstract

Healthcare is an important medical topic in recent years. In this study, the novelty we propose is the intelligent healthcare system using an inequality-type optimization mathematical model with signal-to-noise ratio (SNR) and wavelet-domain low-frequency amplitude adjustment techniques to hide patients’ confidential data in their electrocardiogram (ECG) signals. The extraction of the hidden patient information also utilizes the low-frequency amplitude adjustment. The detailed steps of establishing the system are as follows. To integrate confidential patient data into ECG signals, we first propose a nonlinear model to optimize the quality of ECG signals with the embedded patients’ confidential data including patient name, patient birthdate, date of medical treatment, and medical history. Then, we apply Simulated Annealing (SA) to solve the nonlinear model such that the ECG signals with embedded patients’ confidential data have good SNR, good root mean square error (RMSE), and high similarity. In other words, the distortion of the PQRST complexes and the ECG shape caused by the embedded patients’ confidential data is very small, and thus the quality of the embedded ECG signals meets the requirements of physiological diagnostics. In the terminals, one can receive the ECG signals with the embedded patients’ confidential data. In addition, the embedded patients’ confidential data can be received and extracted without the original ECG signals. The experimental results confirm the efficiency that our method maintains a high quality of each ECG signal with the embedded patient confidential data. Moreover, the embedded confidential data shows a good robustness against common attacks.

## 1. Introduction

In recent years, the technology of Internet of Things (IoT), in which sensors are connected to the network to transmit important sensor information, has gradually matured [1,2,3,4,5,6,7]. Moreover, supporting medical treatment by IoT is even more important [8,9]. The electrocardiogram (ECG) represents the electrical activity of the human heart so that the ECG can be used as a reference for the analysis of cardiac pathology and the diagnosis of the cardiovascular system. Therefore, the ECG is a significant piece of bio-information which needs to be protected and transmitted in the hospital network, and it is necessary to apply the information hiding technology of ECG to protect patients’ rights and information.

Research on protecting ECG information through watermarking or masking techniques remains an important topic. Kong al. [10] and Engin et al. [11] proposed a method of simple data masking of ECG signals, but the method is not blind. Zheng and Qian [12,13] proposed a method to indicate wavelet-domain ECG data in complex non-QRS frames to ensure the recovery of practically undistorted ECG signals. Kuar et al. [14] used a blind masking method to present the secure transmission of ECG signals in wireless networks. Ibaida et al. [15] improved the LSB (Least Significant Bit) watermarking technique and applied this upgraded approach to embed health information into ECG signals. Ibaida et al. [16] announced a watermarking approach to hide patient biomedical information in ECG signals to ensure the integrity of the patient–ECG connection, which is convenient for a wearable health monitoring platform. Nonetheless, it is hard to select the embedding site.

Authors in [17,18] apply a discrete wavelet transform (DWT) with seven-level decomposition to transform the ECG signal and combine the synchronization code with a watermark embedded in the low-frequency sub-band of level 7 to get better Signal-to-Noise ratio (SNR) and bit error rate (BER). However, the quality of all watermarked ECG signals degrades when increasing the embedding strength. Furthermore, Guo and Zhou [18] proposed a model with blind detection by single-channel electromyography. Dey et al. [19] designed watermarks by reversible binary bits and embedded the watermarks into the PPG signal and extracted them based on an error prediction algorithm. Dey et al. [20] embedded the binary watermarked image into the ECG signal proposed so as to obtain a novel session-based blind watermarking method. However, both methods [19] and [20] have the drawback of not being blind. Ayman and Ibrahim [21] established a wavelet-based information-hiding technique that combines encryption and scrambling to protect patients’ confidential data. To protect patient rights and information, transform-domain single-coefficient quantization is administered to ECG digital watermark encryption technology [22,23]. By this practice, the distortion in the PQRST complexes and the amplitude of ECG signal is very small. Jero et al. [24,25] utilized curvelet transformation to determine coefficients for storing diagnostic information. The novelty in their method is the use of curvelet transformation, suitable selection for locating watermark, and a threshold judgement concept. In [26], the original time-frequency watermarking scheme is realized with a lead-independent beat-to-beat adaptive data container design. The authors demonstrated six wavelets, six encoding bit depth values, and two watermark content types to catch the necessary conditions for the watermarked ECG to fit International Electrotechnical Commission (IEC) performance requirements. Sanivarapu et al. [27] announced a wavelet-based watermarking procedure for the patient information hidden in the ECG as a QR image. First, they converted the 1D ECG signal into a 2D ECG image using the Pan–Tompkins algorithm and used a wavelet transform to decompose the 2D ECG image. Then, they decomposed the wavelet detail coefficient and the QR image using QR decomposition for incorporating the data. The concepts proposed in [28] are single-sample quantification, ECG watermarking, and threshold-based compression, which reduce data size while ensuring patient data confidentiality and authenticity.

Since patients’ confidential data including patient name, patient birthdate, date of medical treatment, and medical history have the right to personal privacy, the hospital and related personnel must respect and protect patients’ confidential data during network transmission or telemedicine in the hospital to prevent leaked, stolen, or even misappropriated data. Accordingly, we design a useful information-hiding technique to embed patients’ confidential data into ECG signals in this study. In other words, we propose a new technique to hide patients’ confidential data in ECG signals. Since ECG has high accuracy requirements, especially PQRST waves [29], we rewrite the signal-to-noise ratio (SNR) and the low-frequency amplitude embedding rule as a performance index and constraint so as to obtain an optimized model for embedding sensitive patients’ confidential data including patient name, patient birthdate, date of medical treatment, and medical history attached with patient bed into ECGs. The optimization model is processed by simulated annealing (SA) algorithms, and the results are applied to incorporate the confidential patient data for satisfying better signal-to-noise ratio (SNR), root mean square error (RMSE), and similarity. Consequently, the distortion of PQRST complexes and ECG amplitude is very low, so that the embedded confidential data can fit the requirements of physiological diagnostics. Through network transmission, the ECG with confidential data embedded can be received at the other end and the confidential data can be extracted without the original ECG. In addition, the embedded confidential data shows a good robustness against common attacks.

The rest of this study is as follows. Section 2 gives a sketch of some preliminary work, including discrete wavelet transform (DWT) and simulated annealing (SA). Section 3 presents the proposed method. Section 4 displays the experimental results. Finally, conclusions are shown in Section 5.

## 2. Preliminaries

In this section, we recall the knowledge we need to use in the later section: discrete wavelet transformation (DWT) and simulated annealing (SA).

### 2.1. Discrete Wavelet Transformation (DWT)

The discrete wavelet transformation using scaling and shifting parameters is defined by
(1)φj,k(t)=2j2hj(2jt−k)
(2)ψj,k(t)=2j2gj(2jt−k)

Moreover, Vj=span{φj,k:k∈ℤ} and Wj=span{ψj,k:k∈ℤ} provide that
(3){0}⊂⋅⋅⋅⊂V1⊂V0⊂V−1⊂⋅⋅⋅⊂L2(ℝ)

From a multi-resolution analysis of L2(ℝ) and the subspaces ⋅⋅⋅,W1,W0,W−1,⋅⋅⋅ represents the orthogonal differences of the Vk above. The orthogonal relations give the existence of sequences h={hk}k∈ℤ and g={gk}k∈ℤ which conform to
(4)hk=〈φ0,0,φ−1,k〉 and φ(t)=2∑k∈ℤhkφ(2t−k)
(5)gk=〈ψ0,0,φ−1,k〉 and ψ(t)=2∑k∈ℤgkφ(2t−k)
where h={hk}k∈ℤ and g={gk}k∈ℤ are low-pass and high-pass filters, respectively. In the following work, the host digital audio signal S(n), n∈ℕ represents the sampling of the original audio signal S(t) at the nth sampling time, and the orthogonal Haar wavelet basis is utilized to realize the DWT of the host digital audio signal S(n) by filter bank [30,31].

### 2.2. Simulated Annealing (SA)

Simulated annealing (SA) is an artificial intelligence algorithm that utilizes a random way to approximate the global optimal value of a given function. The SA algorithm is derived from the principle of solid annealing. The solid is heated to a sufficient temperature, and then slowly cooled. During heating, the internal particles of the solid become disordered as the temperature rises, and the internal energy increases., but gradually tend to order, reach equilibrium at each temperature, and finally reach the ground state at room temperature. Then the internal energy is reduced to a minimum. According to the Metropolis criterion, the probability that a particle tends to equilibrium at temperature T is e − ΔE/(kT), where E is the internal energy at temperature T, ΔE is the change number, and k is the Boltzmann constant. Using solid annealing to simulate the combinatorial optimization problem, the internal energy E is simulated as the objective function value f, and the temperature T is evolved into the control parameter t, that is, the simulated annealing algorithm for solving the combinatorial optimization problem is obtained: starting from the initial solution i and the initial value of the control parameter t, the iteration of “generate new solution calculate difference of objective function accept or discard” is repeated for the current solution, the t value gradually decays, and the current solution at the end of the algorithm is the approximate optimal solution obtained. The annealing process is controlled by the cooling schedule (Cooling Schedule), including the initial value t of the control parameters and its decay factor Δt, the number of iterations at each t value, and the stopping condition [32]. The implementation of the SA algorithm is easy due to its simple concept and calculation.

## 3. Proposed Method

The electrocardiogram wave of electrocardiogram (ECG) signals was named by the Dutch physiologist W. Einthoven (the inventor of the ECG). He classified one cardiac cycle into P, Q, R, S, and T complex waves, which are shown in the ECG pattern at the top of Figure 1. Due to the fact that the ECG diagnosis is dependent on the PQRST waves [19], we have to avoid the shape distortion of these waveforms when we add patients’ confidential data including patient name, patient birthdate, date of medical treatment, and medical history into ECG signals. Accordingly, we propose an optimization model to maximize the quality of the embedded ECG signals under low-frequency amplitude modification. As the flowchart shows in Figure 1, we give the details of patients’ confidential data embedding in the following subsections. 

### 3.1. Perform DWT and Binary Bits

Let *S* denote an ECG signal, and cut *S* into several segments. Without generality, each segment has the same length *n* sample points. We then perform DWT decomposition on each segment $S=\left\{s_{1},s_{2},\cdot \cdot \cdot ,s_{n}\right\}$ to get high-frequency sub-band *S_j,H_* and low-frequency sub-band *S_j,L_* in different level *j* = 1,2,3,…, as shown in Figure 2. At the same time, patients’ confidential data including patient name, patient birthdate, date of medical treatment, and medical history is converted to be the binary bits $B=\left\{b_{i}|b_{i}=1\;\mathrm{or}\;0\right\}$ since binary bits *B* can be easily hidden into the lowest-frequency sub-band of DWT. 

### 3.2. Proposed Patient Confidential Data Hiding Technique

Since the energy and quality of a signal are concentrated in the low frequency, the high frequency is susceptible to noise interference and can be easily removed or filtered without affecting the quality of the signal. Therefore, the binary bits are usually embedded in low frequencies to prevent them from being removed, filtered out, or interfered with by noise. In order to define a better threshold when embedding and extracting the binary bits in low frequencies, original audio signal of length L is firstly segmented by I frames, and then k-level DFT is realized on each frame. Therefore, the total number of lowest-frequency coefficients in each frame is $n=L/(I\cdot 2^{k})$ and the mean of these coefficients is
(6)$$m=\frac{1}{n}\sum_{i=0}^{n}\left|c_{i}\right|$$

Then, two thresholds are defined by
(7)$$h_{1}=m+\varepsilon$$
and
(8)$$h_{0}=m-\varepsilon$$
where ε>0 is the embedding strength. Finally, the binary bit of value 0 or 1 is embedded into low frequencies by the following rules.
(9)$$\sum_{i=0}^{n}\left|{\hat{c}}_{i}\right|\geq h_{1},\;\mathrm{if}\;b_{i}=1$$
(10)$$\sum_{i=0}^{n}\left|{\hat{c}}_{i}\right|<h_{0},\;\mathrm{if}\;b_{i}=0$$
where ${\hat{c}}_{i}$ is the embedded DWT low-frequency coefficient corresponding to original DWT coefficient $c_{i}$.

### 3.3. Enhance Performance by the Proposed Optimization Model

Generally, the shape of an ECG signal is distorted when embedding patient confidential data. In order to lessen the distortion of the ECG shape, we take the maximum of the SNR which is defined by
(11)SNR=−10log[∑i=1n(s^i−si)2∑i=1nsi2]
where $\left\{s_{i}\right\}$ means the original ECG signal sample points and $\left\{{\hat{s}}_{i}\right\}$ means the unknown embedded (or modified) ECG signal sample points. Since we realize the DWT using orthogonal wavelet bases, the SNR can be rewritten as
(12)SNR=−10log[∑i=1n(|c^i|−|ci|)2∑i=1n|ci|2]

From the perspective of SNR maximization, we plan to evaluate the unidentified values of $\left\{c_{i}|1\leq i\leq n\right\}$ using the following optimization models.

If the bit $b_{i}=1$ is embedded into $\left\{c_{i}|1\leq i\leq n\right\}$, $\sum_{i=1}^{n}\left|c_{i}\right|$ is modified to $\sum_{i=1}^{n}\left|{\hat{c}}_{i}\right|$ by
(13a)maximize−10log[∑i=1n(|c^i|−|ci|)2∑i=1n|ci|2]
(13b)$$\mathrm{subject}\;\mathrm{to}\;\sum_{i=0}^{n}\left|{\hat{c}}_{i}\right|\geq h_{1}$$If the bit $b_{i}=1$ is embedded into $\left\{c_{i}|1\leq i\leq n\right\}$, $\sum_{i=1}^{n}\left|c_{i}\right|$ is modified to $\sum_{i=1}^{n}\left|{\hat{c}}_{i}\right|$ by
(13c)maximize−10log[∑i=1n(|c^i|−|ci|)2∑i=1n|ci|2]
(13d)$$\mathrm{subject}\;\mathrm{to}\;\sum_{i=0}^{n}\left|{\hat{c}}_{i}\right|<h_{0}$$

### 3.4. Solving the Proposed Optimization Model by SIMULATED Annealing (SA)

Since Equations (13a)–(13d) form an optimization model, we apply the Simulated Annealing (SA) solver to approximately find the optimal solutions of $\left\{{\hat{c}}_{i}|1\leq i\leq n\right\}$. At each point in time, SA stochastically selects a solution that is in sight of the current solution, measures its quality, and then decides to move toward it or to continue with the current solution according to either one of two probabilities between which it chooses on the basis of the fact that the new solution is better or worse than the current one. During the search for the optimal solution, the temperature gradually drops from an initial positive number to zero and affects both probabilities: at each step, the probability of moving toward a better new solution is either kept to 1 or is altered towards a positive value; on the other hand, the probability of moving to a worse new solution is progressively altered towards zero. In this section, SA is applied to approximate the optimal solution of the proposed optimization model. The clear process of SA in solving our proposed optimization model will be listed in the following steps:

Step 1. Given the initial value of parameters, including initial solution *C*_0_, initial temperature *T*_0_, final temperature *T_f_*, cooling rate *r*, and iteration number *D* for each temperature, $C^{\prime \prime }=\left\{\left|{c^{\prime \prime }}_{1}|,|{c^{\prime \prime }}_{2}|,\cdot \cdot \cdot ,|{c^{\prime \prime }}_{n}\right|\right\}$, where $C^{\prime \prime }=\left\{\left|{c^{\prime \prime }}_{1}|,|{c^{\prime \prime }}_{2}|,\cdot \cdot \cdot ,|{c^{\prime \prime }}_{n}\right|\right\}$.

Step 2. For each *d=1…, D* at a temperature *T*, repeat the following:

(1) Randomly generate a new solution $C^{\prime \prime }=\left\{\left|{c^{\prime \prime }}_{1}|,|{c^{\prime \prime }}_{2}|,\cdot \cdot \cdot ,|{c^{\prime \prime }}_{n}\right|\right\}$ and evaluate the difference ΔE=E(C″)−E(C′)={−10log[∑i=1n(|c″i|−|ci|)2∑i=1n|ci|2]−(−10log[∑i=1n(|c′i|−|ci|)2∑i=1n|ci|2])} between current solution $C^{\prime }=\left\{\left|{c^{\prime }}_{1}|,|{c^{\prime }}_{2}|,\cdot \cdot \cdot ,|{c^{\prime }}_{n}\right|\right\}$ and the new solution (neighbors) $C^{\prime \prime }=\left\{\left|{c^{\prime \prime }}_{1}|,|{c^{\prime \prime }}_{2}|,\cdot \cdot \cdot ,|{c^{\prime \prime }}_{n}\right|\right\}$.

(2) The probability of going from the current solution to a new solution (neighbors) is given by an acceptance probability function P(ΔE,T) that depends on ΔE and *T*.
$$P\left(\Delta E,T\right)=\{\begin{array}{cc} 1, & if\;\Delta E\leq 0\\ e^{\left(\frac{-\Delta E}{T}\right)}, & if\;\Delta E>0 \end{array}$$

In case ΔE≤0, the probability function P(ΔE,T) is equal to 1, indicating that the current solution *S* is replaced by the new solution *S’*. In case ΔE>0, the current solution *S* is replaced by the new solution *S’* when the probability function $P\left(\Delta E,T\right)=e^{\left(\frac{-\Delta E}{T}\right)}$ is bigger than a threshold $\hat{S}$.

Step 3. When Step 2 is complete, the temperature *T* is lowered by a cooling rate *r* to a new temperature *T = rT.*


Step 4. Check if the temperature *T* comes to the final temperature *T_f_* to break off simulated annealing. 

After using SA to integrate the essential bodily functions of the patient into the ECG signal *S*, an integrated ECG signal $\hat{S}$ is obtained in each case.

### 3.5. Perform Inverse Discrete Wavelet Transform (IDWT)

After the solution of the optimization model is obtained by SA, the patients’ confidential data is embedded into the lowest-frequency coefficients of DWT of ECGs along with the binary bits. Next, we perform inverse discrete wavelet transform (IDWT), as shown in Figure 3, to obtain the embedded ECGs and transfer the embedded ECGs to end devices over the network. The end devices that receive embedded ECGs use the extraction method in the next section to extract the binary bits to obtain patients’ confidential data. 

### 3.6. Extraction Method

When extracting the patients’ confidential data, we segment the test audio into several frames and then implement DWT on each frame in the same manner as in the embedding process. Suppose *n* consecutive absolute values $\left|{\hat{c}}_{i}^{*}\right|,\;i=1,\;\cdot \cdot \cdot \;,n$ are the optimal-embedded coefficients. We extract the binary bits *B* by using the following rules:If $\sum_{i=0}^{n}\left|{\hat{c}}_{i}\right|\geq h_{1}$, the extracted bit is $b_{i}=1$.If $\sum_{i=0}^{n}\left|{\hat{c}}_{i}\right|<h_{0}$, the extracted bit is $b_{i}=0$.

### 3.7. Architecture of the Proposed Confidential Data Communication System

Figure 4 shows the architecture of the proposed intelligent confidential data communication system. First, we obtain patients Electrocardiogram (ECG) from the ECG sensor module or website in [33,34]. Next, by Section 3.1, Section 3.2, Section 3.3, Section 3.4 and Section 3.5, we first embed the patients’ confidential data including patient name, patient birthdate, date of medical treatment, and medical history attached with patient bed into the ECG signals in the wavelet domain by the proposed hiding method. Then, the embedded ECGs are transmitted to related end devices through internet connections. At the end devices, the optimal-embedded patients’ confidential data are extracted after receiving the embedded ECGs and performing DWT on the embedded ECGs.

## 4. Experiments and Discussion

In experiments, we use the ECG data obtained from the website in [33,34] to simulate the proposed method for each ECG signal with a length of 4096 samples represented by 16-bit. Since the 5-level Haar DWT is performed on the ECG signal, the lowest-frequency sub-band in level 5 has 128 coefficients. The embedding strength ε is set to 2000 and 4000 for *n* = 2 and *n* = 4, respectively. Experimental results and discussion are listed in the following. 

Without generality, the superiority of the proposed method is judged by signal to noise ratio (SNR) and similarity, which are formulated as follows:(14)$$\mathrm{SNR}=-10\log_{10}\left(\frac{\sum_{i=1}^{n}{\left({\hat{s}}_{i}-s_{i}\right)}^{2}}{\sum_{i=1}^{n}s_{i}^{2}}\right)$$
(15)$$\mathrm{Similarity}\left(S,\hat{S}\right)=\frac{\sum_{i=1}^{n}s_{i}{\hat{s}}_{i}}{\sum_{i=1}^{N}{\hat{s}}_{i}^{2}}$$
where $s_{i}$ and ${\hat{s}}_{i}$ denote original ECG signal sample point and embedded ECG signal sample point. The higher the SNR and the similarity, the smaller the distortion of the embedded signal, that is to say, the higher the SNR and the similarity, the better the quality of the embedded signal. 

Our proposed method embeds patients’ confidential data into their ECG signals with minimal distortion and then enables each hidden ECG signal to be of good quality. For example, Figure 3a demonstrates the original ECG signal without patients’ confidential data, and Figure 3b demonstrates the hidden ECG signal with patients’ confidential data using the proposed embedding technique in DWT 5-level decomposition. In Figure 5c,d, the blue curve represents the original ECG signals and the green curve represents the optimal-embedded ECG signals in the case of embedding strength ε = 4000 and *n* = 4. Figure 5e shows that they are almost indistinguishable visually.

Moreover, as shown in Table 1, the main drawback in methods in [22,23,28] is that the SNR, which represents the quality of each embedded ECG signals, decreases significantly with the increase of the embedding strength *Q*. Since SA is applied to optimize the quality of each embedded ECG signal, we improve the disadvantage that the quality of each embedded ECG signal greatly decreases with increasing the embedding strength ε. In other words, our method maintains a good SNR (i.e., good quality) with sufficient hidden capacity for each embedded ECG signal regardless of the increase in the embedding strength ε.

After the embedding, common attacks including re-sampling, low-pass filtering, and noise interference are utilized to test the robustness of the embedded patients’ confidential data. The test methods and average BER results are explained as follows.

(1) Low-pass filtering: Low-pass filter is a kind of signal processing so as to give easy passage to low-frequency signals and difficult passage to high-frequency signals. By the BER information in Table 2, one can find that the robustness of the proposed method against the low-pass filter attack with a cutoff frequency of 3000 Hz and 6000 Hz is significantly higher than the methods in [9,22,23,28].

(2) Noise interference: Gaussian noise is a common noise for audio and other signals. Table 3 lists the experimental results of adding Gaussian noise with various noise intensities to the embedded and compressed audio signal. In common noise intensities −40 dB and −30 dB, the robustness of the proposed method is significantly higher than the methods in [9,22,23,28].

(3) Re-sampling: Resampling converts an audio signal from a given sample rate to a different sample rate. Upsampling or interpolation increases the sampling rate, and downsampling or decimation decreases the sampling rate. Both can be accomplished using integer-valued interpolation or decimation factors. In the proposed method, the embedded-compressed audio signal is first decimated from 44,100 Hz to 22,050 Hz and then interpolated to the original 44100 Hz. This step is repeated two more times from 44,100 Hz to 11,025 Hz and 8000 Hz, then back to 44,100 Hz. The BER during the resampling attack is shown in Table 4. BER information shows that the proposed embedding method leads to lower BER and better robustness.

We performed all experiments above using patients’ Electrocardiogram (ECG) data from the website in [33,34]. With the exception of a few patients with damaged ECGs, most of the ECGs were successfully tested by our proposed method.

## 5. Conclusions

Based on the proposed optimization model and SA algorithm, patients’ confidential data are embedded into ECG signals using DWT lowest-frequency amplitude embedding method. After testing the ECG dataset using the proposed embedding method, the difference between the embedded ECG signal and the original ECG signal is so small that it is almost negligible enough to be suitable for physiological diagnosis. Furthermore, the proposed method improves the disadvantage that the quality of each embedded ECG signal greatly decreases with the increase of the embedded strength ε. On the terminal device, the user can transmit the received embedded ECG through the Internet and then accurately extract the embedded patient’s confidential data by performing DWT on the embedded ECGs received. 

## Figures and Tables

**Figure 1 sensors-22-08341-f001:**
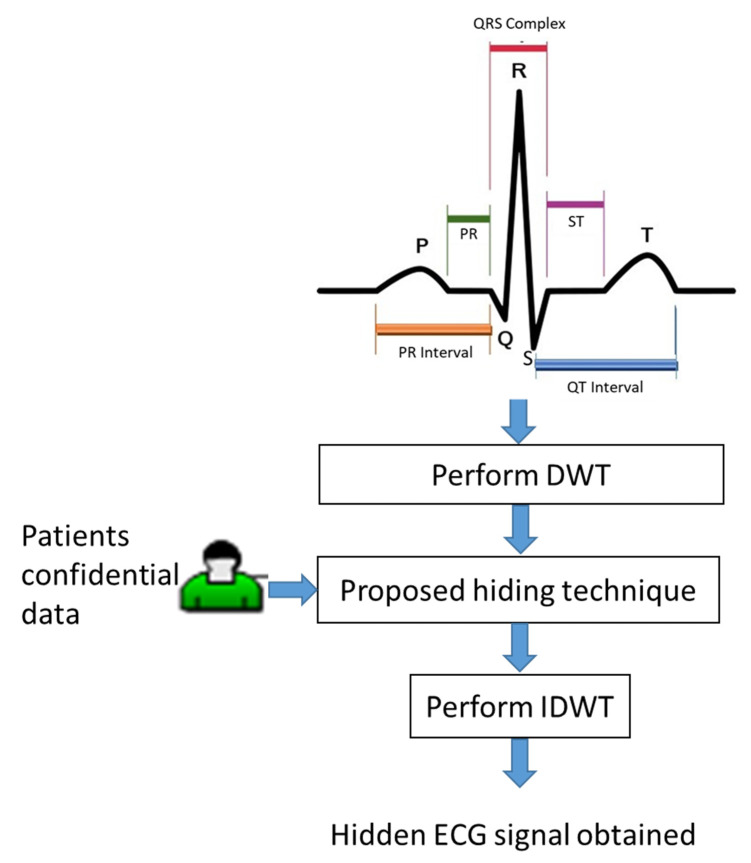
A flowchart of the proposed embedding method.

**Figure 2 sensors-22-08341-f002:**
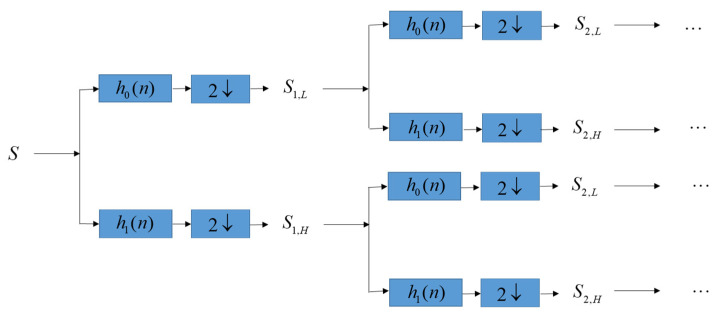
Perform DWT decomposition on each segment $S=\left\{s_{1},s_{2},\cdot \cdot \cdot ,s_{n}\right\}$ to obtain low-frequency sub-band *S_j,L_* and high-frequency sub-band *S_j,H_* in different level *j* = 1,2,3,….

**Figure 3 sensors-22-08341-f003:**
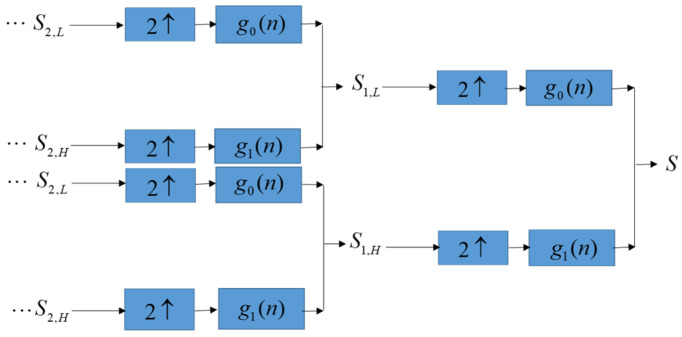
Inverse discrete wavelet transform.

**Figure 4 sensors-22-08341-f004:**
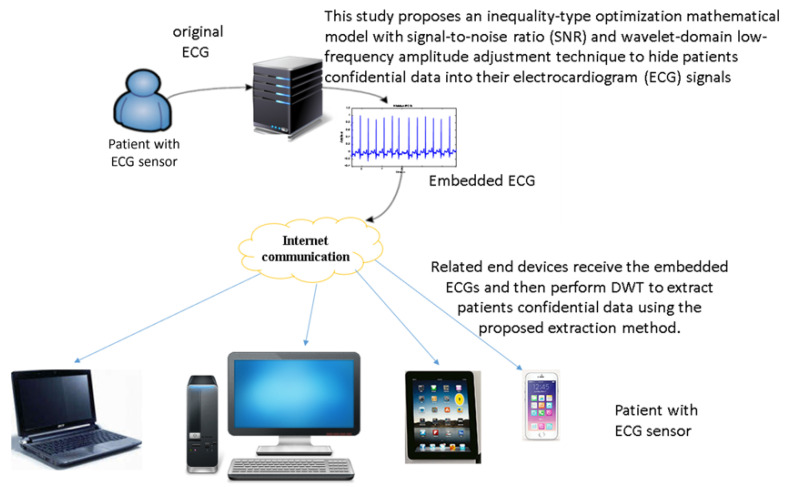
Architecture of the proposed intelligent patients’ confidential data communication system.

**Figure 5 sensors-22-08341-f005:**
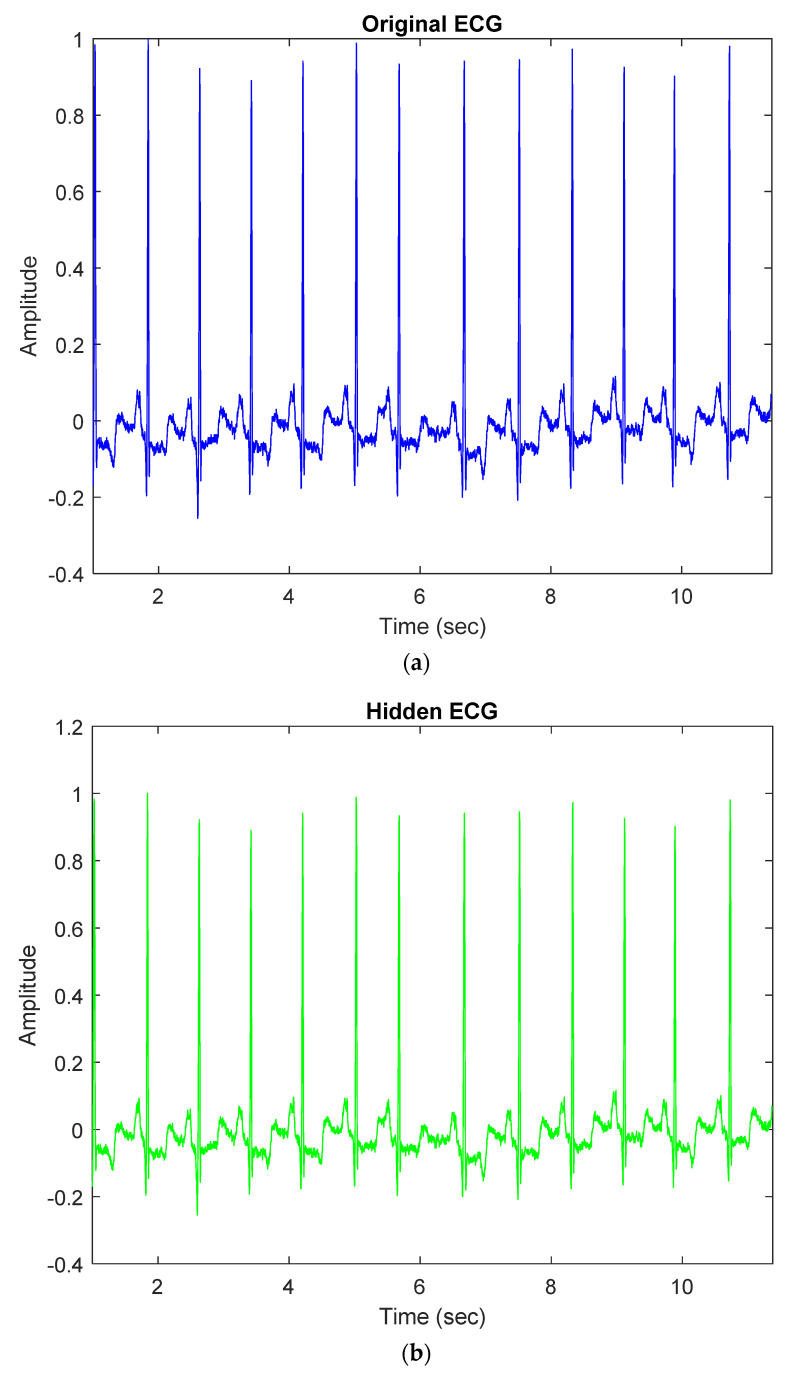
Comparison between original ECG signal and embedded ECG signal for dataset ID 1; (**a**) Original ECG signal; (**b**) Hidden ECG signal; (**c**) Original ECG between 0.09 and 1.09 (s); (**d**) Hidden ECG between 0.09 and 1.09 (s); (**e**) Waveform comparison between 0.09 and 1.09 (s).

**Table 1 sensors-22-08341-t001:** Experimental results of testing amplitude similarity and SNR.

Patient ID	Method	Domain	Embedding Parameters	Amplitude Similarity	SNR
1	Reference [9]	Time domain	*Q* = 500	0.98	41.73
			*Q* = 2000	0.99	41.71
			*Q* = 5000	1	41.72
	Reference [22]	DWT (Level 5)	*Q* = 500	0.99	39.82
			*Q* = 2000	1	36.37
			*Q* = 5000	1	29.15
	Reference [23]	DWT (Level 5)	*Q* = 500	1	40.26
			*Q* = 2000	0.99	35.74
			*Q* = 5000	0.99	29.75
	Reference [28]	Time domain	*Q* = 500	0.99	27.05
			*Q* = 2000	0.99	14.69
			*Q* = 5000	1	8.73
	Proposed	DWT (Level 5)	ε = 2000, *n* = 2	0.99	38.46
			ε = 3000, *n* = 3	0.99	38.44
			ε = 4000, *n* = 4	0.99	38.55
2	Reference [9]	Time domain	*Q* = 500	0.98	38.93
			*Q* = 2000	0.99	38.31
			*Q* = 5000	1	39.24
	Reference [22]	DWT (Level 5)	*Q* = 500	1	40.78
			*Q* = 2000	1	34.67
			*Q* = 5000	1	29.31
	Reference [23]	DWT (Level 5)	*Q* = 500	1	40.26
			*Q* = 2000	1	35.74
			*Q* = 5000	1	29.37
	Reference [28]	Time domain	*Q* = 500	0.99	26.79
			*Q* = 2000	0.99	13.79
			*Q* = 5000	0.99	7.60
	Proposed	DWT (Level 5)	ε = 2000, *n* = 2	1	32.59
			ε = 3000, *n* = 3	1	32.71
			ε = 4000, *n* = 4	0.99	32.29
3	Reference [9]	Time domain	*Q* = 500	0.98	39.24
			*Q* = 2000	0.98	39.72
			*Q* = 5000	0.99	39.70
	Reference [22]	DWT (Level 5)	*Q* = 500	0.99	41.10
			*Q* = 2000	0.99	37.05
			*Q* = 5000	0.99	29.46
	Reference [23]	DWT (Level 5)	*Q* = 500	1	40.26
			*Q* = 2000	1	35.74
			*Q* = 5000	1	28.15
	Reference [28]	Time domain	*Q* = 500	0.98	28.54
			*Q* = 2000	0.99	17.93
			*Q* = 5000	0.99	12.58
	Proposed	DWT (Level 5)	ε = 2000, *n* = 2	1	34.25
			ε = 3000, *n* = 3	1	34.23
			ε = 4000, *n* = 4	1	32.83
4	Reference [9]	Time domain	*Q* = 500	0.99	29.51
			*Q* = 2000	0.99	29.37
			*Q* = 5000	1	29.25
	Reference [22]	DWT (Level 5)	*Q* = 500	1	40.56
			*Q* = 2000	1	37.65
			*Q* = 5000	1	26.05
	Reference [23]	DWT (Level 5)	*Q* = 500	1	40.26
			*Q* = 2000	1	35.74
			*Q* = 5000	1	25.45
	Reference [28]	Time domain	*Q* = 500	0.99	28.05
			*Q* = 2000	0.99	15.72
			*Q* = 5000	0.99	9.78
	Proposed	DWT (Level 5)	ε = 2000, *n* = 2	1	26.21
			ε = 3000, *n* = 3	1	26.95
			ε = 4000, *n* = 4	1	26.78
5	Reference [9]	Time domain	*Q* = 500	0.99	29.25
			*Q* = 2000	0.99	29.41
			*Q* = 5000	0.98	30.00
	Reference [22]	DWT (Level 5)	*Q* = 500	0.99	41.56
			*Q* = 2000	0.99	37.20
			*Q* = 5000	0.99	25.67
	Reference [23]	DWT (Level 5)	*Q* = 500	1	40.26
			*Q* = 2000	1	35.74
			*Q* = 5000	1	25.95
	Reference [28]	Time domain	*Q* = 500	0.99	22.78
			*Q* = 2000	0.99	15.72
			*Q* = 5000	0.98	9.78
	Proposed	DWT (Level 5)	ε = 2000, *n* = 2	0.99	25.16
			ε = 3000, *n* = 3	0.99	25.34
			ε = 4000, *n* = 4	0.99	25.76
6	Reference [9]	Time domain	*Q* = 500	0.99	40.62
			*Q* = 2000	0.99	40.53
			*Q* = 5000	0.98	40.53
	Reference [22]	DWT (Level 5)	*Q* = 500	1	41.39
			*Q* = 2000	1	40.21
			*Q* = 5000	0.99	26.17
	Reference [23]	DWT (Level 5)	*Q* = 500	1	40.96
			*Q* = 2000	1	39.15
			*Q* = 5000	0.99	25.75
	Reference [28]	Time domain	*Q* = 500	0.99	30.71
			*Q* = 2000	0.99	22.34
			*Q* = 5000	0.98	16.89
	Proposed	DWT (Level 5)	ε = 2000, *n* = 2	1	26.83
			ε = 3000, *n* = 3	1	26.39
			ε = 4000, *n* = 4	1	26.56
7	Reference [9]	Time domain	*Q* = 500	0.98	39.24
			*Q* = 2000	0.97	39.16
			*Q* = 5000	0.97	38.56
	Reference [22]	DWT (Level 5)	*Q* = 500	1	40.25
			*Q* = 2000	0.99	30.23
			*Q* = 5000	0.97	28.47
	Reference [23]	DWT (Level 5)	*Q* = 500	1	40.96
			*Q* = 2000	0.98	37.15
			*Q* = 5000	0.98	29.51
	Reference [28]	Time domain	*Q* = 500	0.99	30.71
			*Q* = 2000	0.99	23.54
			*Q* = 5000	0.98	16.59
	Proposed	DWT (Level 5)	ε = 2000, *n* = 2	1	29.83
			ε = 3000, *n* = 3	1	28.39
			ε = 4000, *n* = 4	0.99	28.26
8	Reference [9]	Time domain	*Q* = 500	0.99	38.29
			*Q* = 2000	0.98	37.43
			*Q* = 5000	0.98	37.95
	Reference [22]	DWT (Level 5)	*Q* = 500	1	41.39
			*Q* = 2000	0.98	36.25
			*Q* = 5000	0.99	29.41
	Reference [23]	DWT (Level 5)	*Q* = 500	1	40.96
			*Q* = 2000	0.98	36.41
			*Q* = 5000	0.97	26.35
	Reference [28]	Time domain	*Q* = 500	0.99	31.57
			*Q* = 2000	0.99	26.83
			*Q* = 5000	0.97	17.38
	Proposed	DWT (Level 5)	ε = 2000, *n* = 2	1	28.64
			ε = 3000, *n* = 3	1	27.39
			ε = 4000, *n* = 4	1	28.35
9	Reference [9]	Time domain	*Q* = 500	0.99	35.26
			*Q* = 2000	0.99	34.53
			*Q* = 5000	0.97	34.25
	Reference [22]	DWT (Level 5)	*Q* = 500	1	40.23
			*Q* = 2000	1	32.71
			*Q* = 5000	0.99	25.14
	Reference [23]	DWT (Level 5)	*Q* = 500	1	39.49
			*Q* = 2000	0.99	39.12
			*Q* = 5000	0.99	28.57
	Reference [28]	Time domain	*Q* = 500	0.99	29.67
			*Q* = 2000	0.99	23.54
			*Q* = 5000	0.98	15.28
	Proposed	DWT (Level 5)	ε = 2000, *n* = 2	1	27.38
			ε = 3000, *n* = 3	1	26.93
			ε = 4000, *n* = 4	1	26.65
10	Reference [9]	Time domain	*Q* = 500	1	29.46
			*Q* = 2000	0.99	28.75
			*Q* = 5000	0.98	28.32
	Reference [22]	DWT (Level 5)	*Q* = 500	1	31.39
			*Q* = 2000	1	30.21
			*Q* = 5000	0.96	22.17
	Reference [23]	DWT (Level 5)	*Q* = 500	1	32.96
			*Q* = 2000	1	30.15
			*Q* = 5000	0.97	22.75
	Reference [28]	Time domain	*Q* = 500	0.99	30.71
			*Q* = 2000	0.96	22.34
			*Q* = 5000	0.94	16.89
	Proposed	DWT (Level 5)	ε = 2000, *n* = 2	1	29.83
			ε = 3000, *n* = 3	1	29.39
			ε = 4000, *n* = 4	0.99	29.56
11	Reference [9]	Time domain	*Q* = 500	0.98	27.96
			*Q* = 2000	0.97	27.53
			*Q* = 5000	0.96	27.43
	Reference [22]	DWT (Level 5)	*Q* = 500	1	37.69
			*Q* = 2000	1	32.51
			*Q* = 5000	0.98	21.67
	Reference [23]	DWT (Level 5)	*Q* = 500	1	38.19
			*Q* = 2000	1	33.15
			*Q* = 5000	0.99	22.75
	Reference [28]	Time domain	*Q* = 500	0.99	30.17
			*Q* = 2000	0.99	25.34
			*Q* = 5000	0.98	19.68
	Proposed	DWT (Level 5)	ε = 2000, *n* = 2	1	28.63
			ε = 3000, *n* = 3	1	27.59
			ε = 4000, *n* = 4	1	26.96
12	Reference [9]	Time domain	*Q* = 500	0.98	27.36
			*Q* = 2000	0.98	28.14
			*Q* = 5000	0.99	27.53
	Reference [22]	DWT (Level 5)	*Q* = 500	1	36.49
			*Q* = 2000	0.99	30.81
			*Q* = 5000	0.98	26.21
	Reference [23]	DWT (Level 5)	*Q* = 500	1	37.16
			*Q* = 2000	1	29.15
			*Q* = 5000	0.99	24.75
	Reference [28]	Time domain	*Q* = 500	1	31.87
			*Q* = 2000	0.99	24.39
			*Q* = 5000	0.98	17.68
	Proposed	DWT (Level 5)	ε = 2000, *n* = 2	1	28.93
			ε = 3000, *n* = 3	1	28.49
			ε = 4000, *n* = 4	0.99	28.35

**Table 2 sensors-22-08341-t002:** BER (%) IN THE LOW-PASS FILTERING ATTACK.

	Domain	Cut-Off Frequency (Hz)	BER (%)
Reference [9]	Time domain	3000	27.1
		6000	26.5
Reference [22]	DWT	3000	26.3
		6000	25.4
Reference [23]	DWT	3000	26.5
		6000	24.1
Reference [28]	Time domain	3000	29.7
		6000	28.5
Proposed	DWT	3000	5.62
		6000	2.38

**Table 3 sensors-22-08341-t003:** BER (%) IN THE GAUSSIAN NOISE ATTACK.

	Domain	Amplitude of Noise in dB	BER (%)
Reference [9]	Time domain	−40	18.25
		−30	20.34
		−20	32.67
Reference [22]	DWT	−40	10.09
		−30	13.23
		−20	31.57
Reference [23]	DWT	−40	9.65
		−30	12.76
		−20	32.94
Reference [28]	Time domain	−40	21.83
		−30	24.95
		−20	32.76
Proposed	DWT	−40	1.89
		−30	2.03
		−20	6.75

**Table 4 sensors-22-08341-t004:** BER (%) IN THE RE-SAMPLING ATTACK.

	Domain	Re-Sampling Rate (Hz)	BER (%)
Reference [9]	Time domain	22,050	8.72
		11,025	10.45
		8000	14.92
Reference [22]	DWT	22,050	3.47
		11,025	10.46
		8000	9.65
Reference [23]	DWT	22,050	2.87
		11,025	9.64
		8000	10.52
Reference [28]	Time domain	22,050	12.26
		11,025	15.47
		8000	20.15
Proposed	DWT	22,050	0.72
		11,025	1.23
		8000	2.32

## Data Availability

Not applicable.

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
