# Peer review of "Intelligent Healthcare System Using Mathematical Model and Simulated Annealing to Hide Patients Data in the Low-Frequency Amplitude of ECG Signals"

_sensors, 2022, doi:10.3390/s22218341_

Round 1
Reviewer 1 Report
The manuscript titled “Intelligent Healthcare System Using SA-Based Patients Data in the Low-frequency Amplitude of ECG Signals” is well crafted. This paper suggests a novel technique for encrypting patient data from ECGs. For this purpose, the authors propose to rewrite the signal-to-noise ratio (SNR) and the low-frequency amplitude embedding rule to obtain an optimized model for embedding sensitive patient confidential data. This was done because ECG has high accuracy requirements, especially PQRST waves. The simulated annealing (SA) technique was used to process the optimization model, and the results were then utilized to include private patient information in order to achieve improved signal-to-noise ratio (SNR), root means square error (RMSE), and similarity. Due to the extremely minimal distortion of PQRST complexes and ECG amplitude, the encoded private information was reported to be preserved.
However, I have the following concerns, which should be clarified, and the necessary justification has to be included in the manuscript before the manuscript can be accepted.
· Do not use abbreviations in the manuscript title. Please write the full form of “SA.”
· In the abstract, “SA” should be used in its full form at the first instance.
· In the abstract, the novelty of the work is not clear.
· It is not clear why the patient’s confidential data be embedded into the ECG file. It is up to the operator to embed the data or not. I assume that the authors are proposing this technique for files where already the patient data has already been embedded. This need to be made clear in the introduction section. Also, a proper justification should be included in Figure 4 (summarizing the proposed technology).
· I can not see the patient data in the original signal (Figure 5). I do not understand what type of information the authors are trying to remove. Figures 5a and 5b are exactly the same and have been confirmed in Figure 5c. I do not see any patient data in Figure 5a.
· If you see Table 1, for the method Reference [23], amplitude similarity and SNR are better than the proposed method. Why would someone be interested in the proposed method when better methods are available and that too with lower complexity?
Reviewer 2 Report
The paper presents a new technology to hide patients confidential data in ECGs. Healthcare is a topic of interest to the researchers in the related areas. For the reader, however, a number of points need clarifying and certain statements require further justification. My detailed comments are as follows:
(1)It is suggested that the authors rewrite the abstract section to clarify the purpose of this paper and clearly describe the practical significance of the system.
(2)The content of the relevant research in the first chapter is not comprehensive and complete, and the logic clarity needs to be further improved. It is suggested that the author adjust the structure of this paragraph, and supplement and improve the relevant content.
(3)The ECG signal part in Figure 1 is vague, and it is suggested that the author repaint the flow chart of this part to avoid missing important information.
(4)Is there a problem with the values of the different levels of j mentioned in 3.1? Please ask the author to verify it.
(5)The formula in 3.2 has problems such as the format is not centered. Figure 3 and Figure 4 do not set the correct format. The size of Figure 3 in Figure 5 should be as consistent as possible, and the table format should also be set uniformly.
(6)The experimental data selected in this paper is too single source, and the number of samples is also small. It is suggested that the author experiment on multiple groups of ECG data, and give the corresponding conclusions, to avoid the problem of limited applicable environment.
Round 2
Reviewer 1 Report
The authors have responded to my queries. Hence, the paper may be Accepted